# Impact of Selected Meteorological Factors on COVID-19 Incidence in Southern Finland during 2020–2021

**DOI:** 10.3390/ijerph192013398

**Published:** 2022-10-17

**Authors:** Lisa Haga, Reija Ruuhela, Kari Auranen, Kaisa Lakkala, Anu Heikkilä, Hilppa Gregow

**Affiliations:** 1Finnish Meteorological Institute, Meteorological and Marine Research Programme, Weather and Climate Change Impact Research, P.O. Box 503, 00101 Helsinki, Finland; 2The Center of Statistics, University of Turku, 20500 Turku, Finland; 3Finnish Meteorological Institute, Space and Earth Observation Centre, Earth Observation Research, P.O. Box 503, 00101 Helsinki, Finland; 4Finnish Meteorological Institute, Climate Research Programme, Atmospheric Research Center of Eastern Finland, P.O. Box 503, 00101 Helsinki, Finland

**Keywords:** COVID-19 incidence, absolute humidity, temperature, meteorological factors, distributed lag non-linear model, SARS-CoV-2

## Abstract

We modelled the impact of selected meteorological factors on the daily number of new cases of the coronavirus disease 2019 (COVID-19) at the Hospital District of Helsinki and Uusimaa in southern Finland from August 2020 until May 2021. We applied a DLNM (distributed lag non-linear model) with and without various environmental and non-environmental confounding factors. The relationship between the daily mean temperature or absolute humidity and COVID-19 morbidity shows a non-linear dependency, with increased incidence of COVID-19 at low temperatures between 0 to −10 °C or at low absolute humidity (AH) values below 6 g/m^3^. However, the outcomes need to be interpreted with caution, because the associations found may be valid only for the study period in 2020–2021. Longer study periods are needed to investigate whether severe acute respiratory syndrome coronavirus 2 (SARS-CoV-2) has a seasonal pattern similar such as influenza and other viral respiratory infections. The influence of other non-environmental factors such as various mitigation measures are important to consider in future studies. Knowledge about associations between meteorological factors and COVID-19 can be useful information for policy makers and the education and health sector to predict and prepare for epidemic waves in the coming winters.

## 1. Introduction

In temperate climate regions viral respiratory infections are known to have higher incidence during the winter, and earlier studies have shown that ambient temperature and humidity play a role in the seasonal nature of respiratory viral infection outbreaks [1,2,3]. Data from England and Wales show that coronaviruses predating SARS-CoV-2 had a similar seasonal distribution as influenza A and bocavirus during 2012–2019 [4].

The association between meteorological factors and COVID-19, a disease caused by severe acute respiratory syndrome coronavirus 2 (SARS-CoV-2), have been inconsistent in early epidemiological studies [5]. A recent review found that despite statistical association between meteorological variables and the incidence of other respiratory infections, there was lack of consensus between the 43 included studies on how meteorology modifies the transmission of SARS-CoV-2 [6]. During the early stages of local COVID-19 epidemics, the extent of SARS-CoV-2 transmission depended on government interventions and population behavior rather than meteorological conditions [7].

A number of published studies report associations between meteorological factors and COVID-19. For example, in the United States [8], cold and dry weather as well as low levels of ultraviolet radiation were associated with increased SARS-CoV-2 transmissibility, with humidity playing the largest role. A study of 188 countries showed that in the northern hemisphere, lower than average mean temperature and relative humidity values were associated with a higher relative risk of COVID-19. Higher than average levels of temperature and humidity in turn showed lower risk [9]. Based on short-term data from England, an increased risk of COVID-19 incidence was found at daily ambient mean temperatures of around 11–13 °C which correspond to typical conditions during the coldest months of the year in England [10]. In China, a study showed that the incidence of COVID-19 decreased with increasing temperature [11].

Findings about the role of humidity vary as well, possibly depending on the type of humidity variable used. For instance, Ai et al. [12] found that the way relative humidity had affected COVID-19 prevalence depended on the range of humidity variation in the selected countries. In studies regarding the association between absolute humidity and COVID-19 incidence, Runkle et al. [13] found a positive association with humidity values around 6–9 g/kg in the United States, and Nottmeyer and Sera [10] found the humidity value of 6–8 g/m^3^ to be associated with the highest risk of infection in England. Higher pollen concentrations in the air are also found to correlate with increased infection rates, including in those not allergic to pollens [14]. In Finland, the dependence of the daily incidence of COVID-19 on meteorological factors (temperature, relative humidity, and air pollutants) in different hospital districts was studied in the beginning of the epidemic (1 January through 31 May 2020) but no firm associations were found [15].

Causal pathways between weather or climate and viral epidemics are not fully understood. The World Meteorological Organization (WMO) [5] divided the mechanisms of how meteorological factors can influence transmission into virus viability, host immunity and human behavior. Nichols et al. [4] divided the drivers of seasonal infection dynamics into immunological, weather-related, social, and travel-related factors. Studies on virus survival rates have shown that prolonged virus viability and transmissibility are associated with cooler and drier low-humidity environments as these conditions favor rapid evaporation of water from exhaled aerosols [2,5,16]. Studies on environmental conditions and antiviral defense in human airway epithelium implies a relationship between the host defense systems and changes in temperature and the water content of the air, but high temperature can also impair antibody production [17]. Cold weather increases respiratory symptoms and functional disability especially among patients with asthma and allergic rhinitis as the inhalation of cold air may contribute to increased susceptibility to respiratory infections [17,18]. Weather impacts human behavior also directly as people tend to spend more time indoors during cold, hot or rainy weather, which provides more favorable conditions for virus transmission [5,19].

In Finland, the first case of COVID-19 was detected in Lapland on 21 January 2020, but the actual number of cases started to increase only in late February [20,21]. By mid-March the disease was considered an epidemic and on 16 March 2020, the government announced a state of emergency and imposed physical distancing measures to slow down the spread of the virus [20]. After the first epidemic peak in March–April 2020, two further peaks were noted by 31 May 2021. The second peak occurred in November–December 2020 and the third one occurred in March 2021. Vaccinations against COVID-19 started in 2021 and have mitigated COVID-19 outbreaks and reduced incidence, hospitalizations, and deaths in the population [22].

In this study, we aim to provide further understanding about the impact of meteorological factors on the COVID-19 epidemic in Finland, a country with a northern European climate. Specifically, we aim to model the weather-dependence of COVID-19 for the time-period from August 2020 through May 2021 at the Hospital District of Helsinki and Uusimaa, the extended capital region of the country and assess the role of absolute humidity and temperature in explaining the intensity of the epidemic waves.

## 2. Materials and Methods

### 2.1. Data

The daily numbers of laboratory confirmed COVID-19 cases in Finland were collected from the Finnish Institute for Health and Welfare repository (THL, 2021) for two different time periods: from 27 February 2020 to 31 May 2021, and 1 August 2020 to 31 May 2021. The morbidity dataset does not include information on gender or age. We focused on the Hospital District of Helsinki and Uusimaa (HUS) (Figure 1), as this hospital district is the most populated and exhibited the largest numbers of COVID-19 cases in Finland. The HUS area includes Helsinki and the wider metropolitan area of Finland as well as several other municipalities.

We analyzed the two time periods separately. The main analysis is based on the second period (August 2020–May 2021), because this time series is more homogeneous than the time series including spring 2020: the testing capacity to confirm COVID-19 cases was not sufficient in spring 2020 when the pandemic started rapidly. Furthermore, the first wave peak weakened quickly, quite likely due to the mitigation measures by the government, and the cases started to rise again in autumn 2020 [21]. On the other hand, after the main study period, in summer 2021 the new delta virus variant took over and the share of the vaccinated population increased substantially [23]. Table 1 shows an overview of the COVID-19 data and meteorological factors during the study period.

The meteorological data consisted of daily average values of temperature (T), absolute humidity (AH), air pollution, and ultraviolet radiation (UVR). Absolute humidity (g/m^3^) values were calculated with a formula based on the WMO recommendations (WMO No-8) [24]. For air pollution we used particulate matter PM10 measured at the urban background station in Kallio, maintained by the Helsinki Region Environmental Services Authority. The level of UVR was derived by calculating erythemally weighted UV daily doses [25] from spectral UV irradiance measured by a Brewer spectroradiometer located in Kumpula, Helsinki (60.20 N, 24.96 E) [26,27]. Satellite retrievals were used to fill data gaps, which may have occurred due to either malfunction of an instrument or calibrations, and surface UV radiation products from TROPOMI [28,29] and OMI [30,31] instruments were used for measurements. OMI UVR data were downloaded from NASA Aura data validation center [32].

We characterized the government restriction actions using the Oxford Coronavirus Government Response Tracker (OxCGRT). Data for Finland were derived directly from the site OurWorldData (https://ourworldindata.org/COVID-stringency-index, (accessed on 15 June 2022). The index combines the following nine metrics: school closures; workplace closures; cancellation of public events; restrictions on public gatherings; closures of public transport; stay-at-home requirements; public information campaigns; restrictions on internal movements; and international travel controls [9,33]. The index uses a scale from 0 to 100 to estimate how strict the policy responses were. The stringency index (STRI) data for Finland during the study period is presented in Appendix A.

### 2.2. Statistical Methods

We studied short-term associations between temperature and absolute humidity with the daily number of new COVID-19 cases, using a distributed lag non-linear model (DLNM). DLNM provides a framework to describe exposure–response dependencies involving non-linear and delayed effects of environmental stressors [34,35]. The method is a regression analysis using a generalized linear model with a quasi-Poisson family of distributions. The general model formula is:(1)$$g\left(\mu_{t}\right)=\alpha +s\left(x_{t};\eta \right)+\overset{J}{\underset{i=1}{\sum }}hj\left(c_{ti},\gamma k\right)$$
where *g*() is a monotonic link function, $\mu_{t}$ = *E*(*Yt*), where *E*(*Yt*) is the expected value of the number of COVID-19 cases at day *t*. *α* is the intercept and *s*(*x_t_*;*η*) is the exposure–response function or cross-basis matrix of the selected meteorological predictor *x_t_* (temperature or absolute humidity). The confounding variables are presented with *c_ti_* defined with function *hj* with parameter vector *γk*. We used a cross-basis function to create a two-dimensional matrix of AH (daily mean absolute humidity) or T (daily mean temperature), with natural cubic splines to account for the time lagged effect of the meteorological predictor in question. The Akaike Information Criteria (AIC) were used to select the number of knots for absolute humidity (or temperature) and the time lags. In addition to knots selection by AIC, we fitted an additional model with three predefined knots for the exposure variables and one knot for lags. The knots were equally spaced within the range of the predictor while the positions of the lag knots were based on equally spaced log values.

The relative risk was studied with two different models: a simple model (Model 1) with the predictors T or AH and day-of-the-week effect (dow) as a confounding factor, and a complex model (Model 2) with several other confounding factors (see Model 2). As T and AH are highly correlated, they could not be used in the same model. The results are reported as incidence rate ratios (relative risks, RR) that summarize the associations of interest [36,37].

Model 1: a simple model
(2)$$g\left(\mu_{t}\right)=\alpha +s\left(x_{t};\eta \right)+dow$$

Model 2: a complex model with several confounding factors
(3)$$g\left(\mu_{t}\right)=\alpha +s\left(x_{t};\eta \right)+dow+ns\left(yday,\;df\right)+\mathrm{PM}10+UV+STRI$$

*dow* is a dummy variable for the day of the week on day *t*,

*yday* is the natural cubic spline function for the day of the year with degrees of freedom (*df* = 3),

PM10 is respirable particles of diameter 10 micrometres or smaller (µg/m^3^),

*UV* is the erythemally weighted *UV* daily dose,

*STRI* Government Response Stringency Index for Finland.

We estimated several alternative models using different lag durations (7, 14 and 21 days). In previous COVID-19 studies, a lag of 14 days has often been used as the maximum lag, but there are inconsistent results on which lag duration is associated with COVID-19 [36]. The incubation period for COVID-19 is on average 5 days but to investigate delayed effects such as secondary infections also longer lag of 14 or 21 days were studied. For T and AH, we used 5 °C and 7 g/m^3^ as reference point values as these values were close to the predictor mean in the observed meteorological time series. Dow is a factor to control the effect of weekdays. Seasonal trends were added into Model 2 with a time controlling term, day of the year (yday) with 3 df. All incidence risk ratios or RR were estimated with 95% confidence intervals (CI).

We present results with 14-day and 21-day lags and, as a sensitivity analysis, also provide results with 7-day lag in the Appendix A. All analyses were performed separately for the two time periods. The main analysis covers the period from August 2020 to May 2021. Results covering the entire period from February 2020 to May 2021 are provided in the Appendix A.

The statistical analyses were performed with R 4.1.2 package “dlnm” [36].

## 3. Results

### 3.1. Epidemic Waves in the HUS Area

Figure 2 shows the time series of daily COVID-19 cases with the daily values of absolute humidity and mean temperature during the study period. The highest number of new daily COVID-19 cases occurred during the third wave, which started in February 2021 and peaked in mid-March with a subsequent decline in April. The dominant virus variant for the second and third epidemic peak in Finland was the alpha (B.1.1.17) variant, but also other virus variants such as the beta (B.1.351) variant were present in the third peak [38,39]. Two weeks prior to and during the third wave the daily AH averages were between 1 and 5.8 g/m^3^ and the daily average T between −14.7 and +4.8 °C. During the second wave (November–December 2020), the daily numbers of cases were lower than during the third wave. The AH varied during the second epidemic peak between 3.0 and 8.7 g/m^3^ and the daily average T between −1.5 °C and +10.6 °C.

### 3.2. Relative Risks of COVID-19 by Absolute Humidity and Temperature

Figure 3 shows the relative risk (RR) of COVID-19 estimated by Models 1 and 2 and using AH or T as the exposure factor over a 14-day lag with two different knot alternatives. We first subjectively fixed the knots to three for the exposure variables (absolute humidity and temperature) and one for the lag. Alternatively, the number of knots was allowed to vary according to the lowest AIC but was limited to a maximum of six knots for exposure and one or two knots for lag.

Model 1 showed a higher RR with AH values below 6 g/m^3^ and at 9–10 g/m^3^ when compared to the reference value of 7 g/m^3^. By contrast, high AH values (>11 g/m^3^) were associated with a smaller RR for both knot options. In model 1 with T as the exposure factor, the RR was highest at around −7 to −10 °C, compared to the reference value of T = +5 °C. The risk decreased with temperatures higher than +10 °C.

In Model 2, which included several confounding factors, the overall RR values for AH were similar to those in Model 1 as high AH values (>11 g/m^3^) were associated with lower COVID-19 risk for both knot options. The highest RR was found with AH less than 6 g/m^3^ and between 8–10 g/m^3^. For temperature, the highest RR values in Model 2 were found between −0 and −10 °C.

Several confounding factors, such as particulate matter, ultraviolet radiation and the stringency index, modified the exposure–response relationships, but the pattern remained fairly similar to the one in the simpler Model 1. The absolute humidity and lag associations showed varied results. For instance, the highest RR values were found between lag 0 to lag 10 in both models when the number of knots was based on AIC, and for the fixed three knots option, the RR was elevated from all lags up to 14 days with low AH values. For temperature, the RR and lag associations showed mixed results. In Model 1 with T as the exposure, the RR was highest between lags 8 to 14 while for Model 2, the RR was elevated for all lag days up to 14 days. We did not investigate the impact of each individual confounding factor separately.

### 3.3. Model 2 Analysis with Different Lags

The role of the delayed effects of meteorological factors on the COVID-19 incidence was studied over 21-day and 14-day lag periods to investigate whether their differences occur when varying the length of the total of lag days used. The two different lag time lengths were applied for both absolute humidity and temperature in Model 2. The results are presented in Figure 4.

Increased RR was found at low humidity values varying from approximately 1.1 to 6.0 g/m^3^, but also at higher humidity values between 8 to 10 g/m^3^, as compared to the reference AH value of 7 g/m^3^ (Figure 4). At low absolute humidity values the RR was elevated up to a 14-day lag. The RR for temperature showed the highest risk between 0 °C and −10 °C, compared to the reference T value of +5 °C, and the risk was elevated with all lags up to day 21 for both knot options.

The delayed impact of meteorological factors on COVID-19 is not clear. For low absolute humidity values the risk lasted for 14 days but with other AH values the results were mixed. Similarly, for low temperatures the risk remained elevated for up to 21 days.

### 3.4. Overall Cumulative Exposure–Response

Figure 5 presents the overall cumulative associations (net risks) of COVID-19 across all 14-day lags for AH and T for Model 2. In dry conditions the overall RR increases substantially. For instance, at AH = 2 g/m^3^ the overall RR was 1.73 (95%-CI: 1.19; 2.51) with the reference value AH = 7 g/m^3^. Similarly, the RR at T = −7 °C was 3.95 (95%-CI: 2.88; 5.423) with the reference value T = +5 °C.

Figure 6 presents predictor-specific RR and lag associations. The figure shows RR at AH = 2 g/m^3^ and T = −7 °C over a 14-day lag, with Model 2 with three fixed knots. The RR stays elevated in both figures over the 14-day lag.

As a sensitivity analysis, we show additional modelled relationships (see Appendix A), and show that the different model versions as well as longer study period including spring 2020 produced similar generic patterns, i.e., increased COVID-19 risks at low values of AH or T, and decreased risks at higher values of AH or T.

## 4. Discussion

We modelled the effect of ambient temperature and absolute humidity on the incidence of COVID-19 cases at the Helsinki and Uusimaa Hospital District (HUS) area by applying DLNM and using different modelling versions. In the simple model (Model 1), only one meteorological factor at a time was used as the independent variable while the complex model version (Model 2) included several environmental and non-environmental confounding factors. In addition, model versions with different lag lengths and different numbers of knots were used to study how consistent the modelled relationships were. Our results showed that the overall patterns of the exposure–response relationships were similar across different models, although the details of the relationships differed. Our study indicated that a higher incidence of COVID-19 was associated with low absolute humidity and low ambient temperature which coincide with the third epidemic wave peak in February–March 2021. It is likely that Model 2 explains the associations between meteorological factors and COVID-19 better than Model 1 as it also includes non-environmental confounding factors.

Studies on the role of meteorological factors on the spread of COVID-19 have showed varied results in the published literature. Nevertheless, several studies suggest that it is likely that weather contributes to the better survival of the virus in the winter months. Our results from the 10-month study period from August 2020 to May 2021 agree with the notion that viruses tend to survive longer in colder and drier conditions (e.g., [4,17,40]). When comparing to the results of Nottmeyer et al. [41], we consider that it is also important to perform national level modelling since global multi-country studies may not adequately describe in detail the situation and climatic conditions in Finland. Trends in meteorological factors such as temperature or humidity can increase or decrease the activity of a specific virus at certain times of the year [4,42,43]. In this study we did not study in detail meteorological factors such as UVR, but studies suggest also that solar radiation is highly effective in inactivating SARS-CoV-2 [44,45]. UVR, PM10 and the stringency index were used as confounding factors in the regression model. However, non-meteorological factors such as host resistance and human behavioral changes also contribute to the seasonality of virus infections. It is worth noting that after the time period covered in this study, a new delta (B.1.617) variant started spread in Finland and led to increased COVID-19 incidence during the summer months in 2021, but the largest increase in the incidence occurred during the winter 2022 when omicron (BA-variants) started to spread [23]. Thus, with new variants, SARS-CoV-2 can spread effectively throughout the year and further studies are needed to elaborate on the influence of meteorological factors.

Due to the relatively short study period, any conclusions need to be made with care. The modelled exposure–response relationships might indicate true causal pathways but could also be only due to coincidence. The highest incidence of COVID-19 occurred during the third wave of the epidemic in March 2021 (Figure 1), when the weather conditions were colder and drier than during the second wave in November–December 2020. Consequently, the exposure–response relationship in Model 2 also showed an increased RR at higher humidity values ranging from 8 to 10 g/m^3^, conditions that prevailed during the second epidemic wave (Figure 3). The delayed impact of meteorological factors on COVID-19 is not clear. For low absolute humidity values the risk varied between the knot options and models demonstrated elevated risk up to 14 days. Similarly, for specific low temperatures, the risk remained elevated for up to 21 days. Based on our results, it might be useful to consider weather conditions acting as a protective factor as well. According to our study, AH values of more than 10 g/m^3^ and daily mean temperature values higher than 10 °C showed a decreased relative risk for COVID-19 daily incidence.

Starting from March 2020 pandemic mitigation measures were present at the Helsinki and Uusimaa Hospital District; social distancing and working remotely were mandatory or highly recommended, meaning that the study period does not represent the pre-pandemic normal conditions when people work, travel and can meet each other flexibly. Results based on the state of emergency conditions set up during the coronavirus pandemic are very different to conditions in earlier studies on the seasonality of respiratory viral infections and thus the influence of weather factors is probably not yet clearly seen or may give misleading results. Furthermore, the coronavirus infection rates in Finland were most likely slowed down during social distancing orders, which were present during the study period [21]. To study seasonality, we would need several years of data while keeping in mind that other non-environmental conditions have also changed from 2020–2021. We used the time series from August 2020 until May 2021 as this study period presents a sufficiently homogeneous time series when considering non-environmental background factors such as vaccines and mitigation measures. During the study period from August 2020 to May 2021 most of the population was not vaccinated, and also the policies and recommendations changed substantially after May 2021 [23]. However, we consider that it is important to conduct later further studies with longer time series in order to better understand the role of different variants and mitigation measures on the weather dependence of COVID-19.

In addition to the short time period and the presence of social distancing, there are several challenges that make COVID-19 modelling with meteorological factors difficult. The DLNM methodology applied in this study is a good approach to research short-term effects with different lags. However, the ways in which the analysis and selection of modelling variables such as the reference points or knots were performed means that the comparison difficult and even analyses based on data from the same country can provide very different results [6]. Furthermore, significant differences could not be seen in our results from the two knot selection alternatives. The models with a fixed number of three knots may yield a simplified association but can also lead to model underfitting. On the other hand, model overfitting may be a consequence in a model version where knot selection is based on AIC, especially if the maximum number of knots is not limited.

To overcome some of the above challenges in future studies, investigating multiple regions would be useful, but preliminary tests for other hospital districts in Finland have proved to be challenging due to the low number of daily cases of COVID-19. Furthermore, based on other existing research, the use of the effective reproduction number instead of the daily number of new COVID-19 cases could be studied [6]. Modelling the seasonality of other common viral respiratory infections in Finland with the DLNM modelling framework would also be useful to understand the influence of meteorological factors. In this study, the stringency index was selected to describe the extent of governmental mitigation measures, but an evaluation of how well this index fits Finland has not been performed. The index data has been available since 1 January 2020 and it has been used as an epidemiological index in several COVID-19 studies [10,12,41,46,47]. The index shows how the mitigation measures changed during the study period.

One possibility for further studies could be to review other investigations conducted for the same period in the Uusimaa region. For instance, weather conditions explain heating and cooling demands, which impact indoor humidity which again depend on the ventilation systems in use. Data from indoor and outdoor investigations, such as from the study started with the Finnish Football Association by Gregow et al. [48], could additionally be used as study material. A lot more research is needed on chains of impact, but in the future the outcomes could be used in the development of multi-hazard early warning systems (MHEWS) for infectious diseases. These systems are explained, for instance, by Rogers et al. [49].

Despite the uncertainties, for the upcoming cold seasons it might be worthwhile to apply the associations of COVID-19 incidence with low temperature and absolute humidity from our study period in 2020–2021 as information for the HUS area. The outcomes could be used in epidemic models, biometeorological forecasts, multi-hazard early warning systems or as information to health authorities, education sector and healthcare providers.

## 5. Conclusions

Our study from the HUS area in southern Finland in 2020–2021 found a non-linear association between the selected meteorological variables and COVID-19, linked with the second and third epidemic waves. The highest relative risk was found to correlate with low temperature and humidity values, while high temperature and humidity values were associated with lower RR. However, the results need to be interpreted with caution because of the short study period and the efficient government policies and recommendations that took place during the early years of COVID-19.

Seasonality will likely affect the incidence of COVID-19 in Finland in the future, but longer study periods with relevant non-environmental influences are needed to confirm the associations between COVID-19 and meteorological factors. The associations between meteorological factors and COVID-19 can be useful information for healthcare professionals to predict and prepare for epidemic waves in the coming years.

## Figures and Tables

**Figure 1 ijerph-19-13398-f001:**
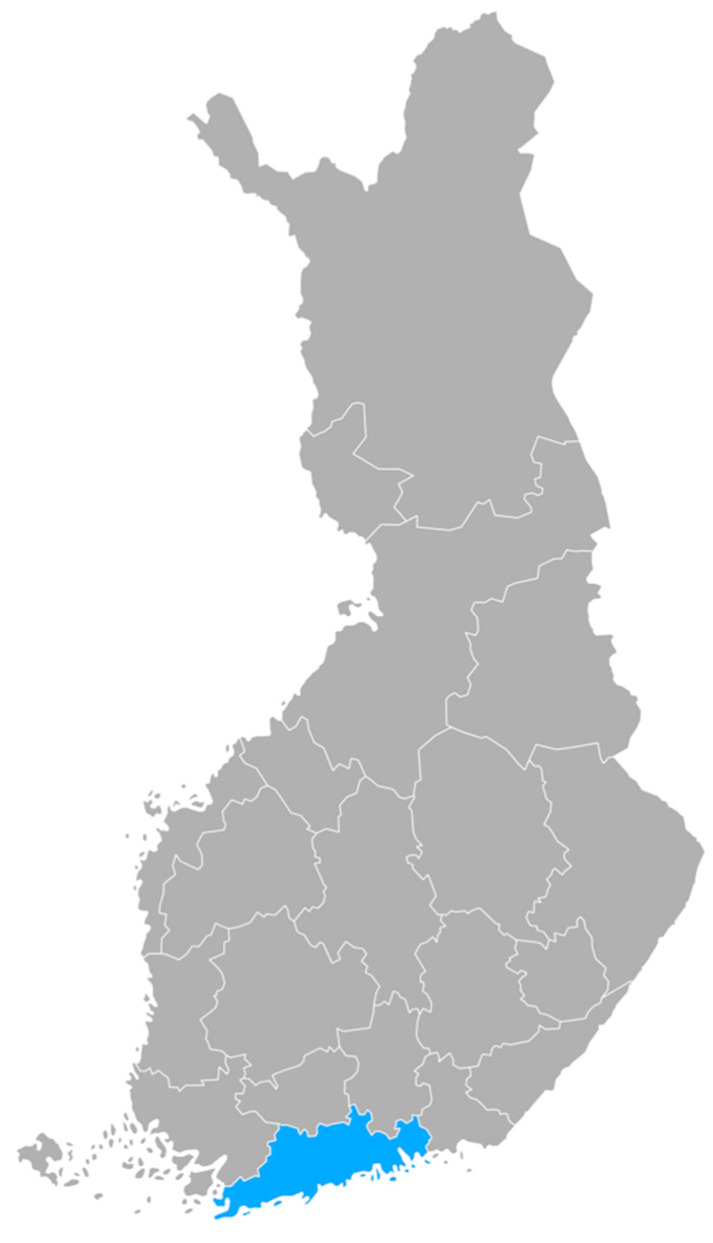
The map shows the 21 different hospital districts in Finland. The study area, the Hospital District of Helsinki and Uusimaa (HUS), is shown in blue.

**Figure 2 ijerph-19-13398-f002:**
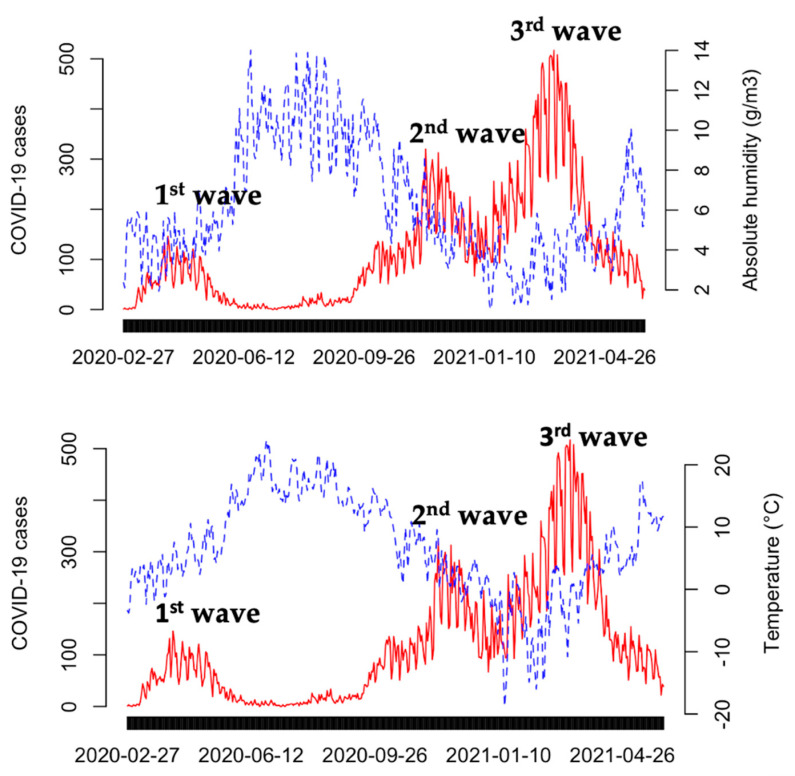
Daily average absolute humidity values (**upper figure**) and daily mean temperature (**lower figure**) with daily numbers of laboratory-confirmed COVID-19 cases (red color) in the Helsinki-Uusimaa Hospital District (HUS). Three epidemic wave peaks took place during the study period from 27 February 2020 to 31 May 2021.

**Figure 3 ijerph-19-13398-f003:**
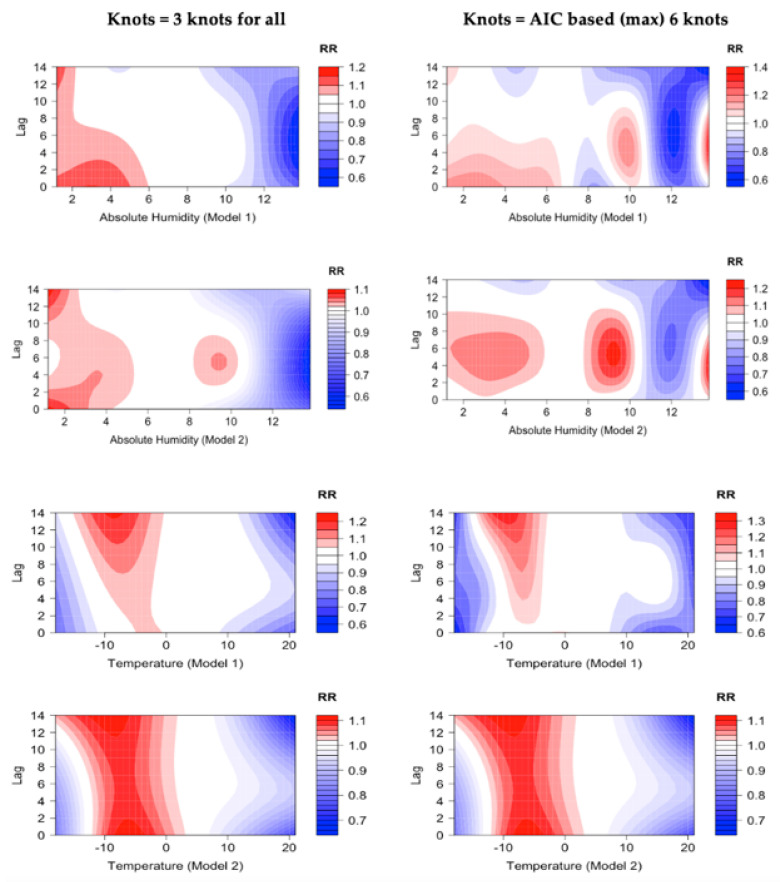
Relative risks of COVID-19 by absolute humidity, daily mean temperature, and time lag, compared to the reference values of 7 g/m^3^ and 5 °C. The figures present RR for Models 1 and 2 with up to 14-day lag in the Helsinki-Uusimaa hospital district (HUS) from 1 August 2020 until 31 May 2021. Left panels: the number of knots for the exposure–response function was fixed at three knots. Right panels: the number of knots were based on the lowest AIC with a maximum limit of six knots. Increased RR is presented as red, decreased RR as blue, and white indicates no significant difference.

**Figure 4 ijerph-19-13398-f004:**
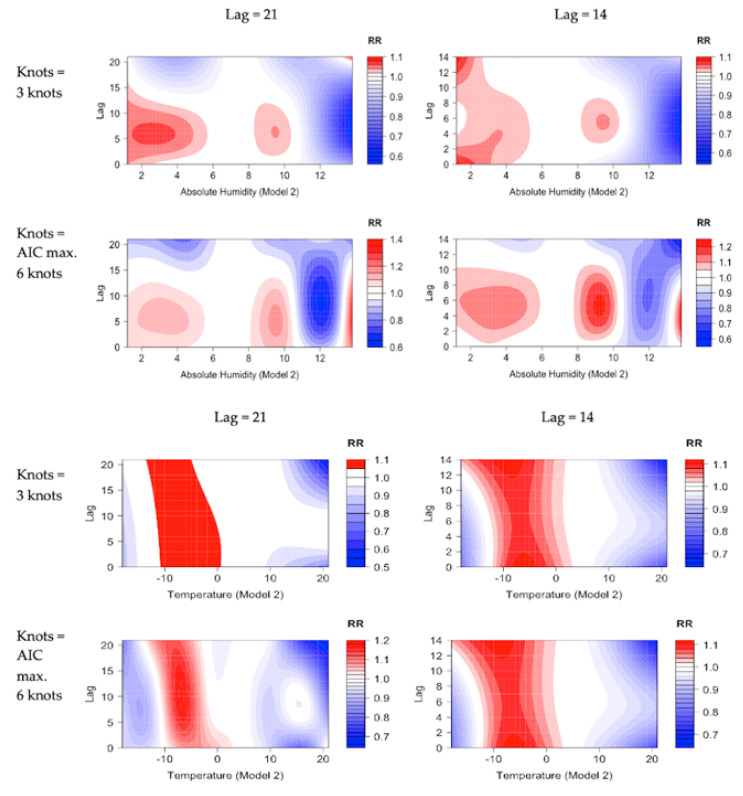
Relative risks of COVID-19 by absolute humidity, temperature, and lag, compared to the reference values of 7 g/m^3^ and +5 °C. The figures present RR for Model 2 with lag up to 21 and 14 days in the Helsinki-Uusimaa hospital district (HUS) for the time period 1 August 2020 until 31 May 2021. The results were calculated for both a fixed number of three knots and using AIC with a maximum of six knots. Increased relative risk is presented in red, decreased risk in blue, and white indicates no significant difference.

**Figure 5 ijerph-19-13398-f005:**
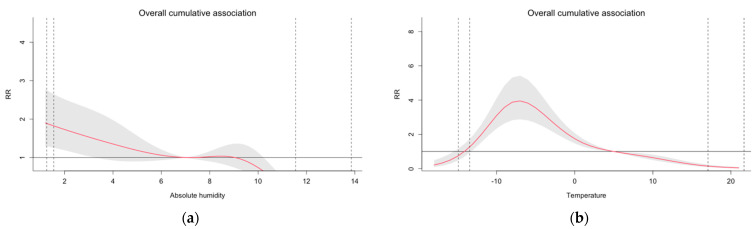
Overall associations between (**a**) absolute humidity and COVID-19 morbidity incidence with the reference value of AH = 7 g/m^3^ and (**b**), temperature and COVID-19 morbidity, with the reference value of T = +5 °C in the Helsinki-Uusimaa Hospital District (HUS) from 1 August 2020 until 31 May 2021. The figures present the overall cumulative RR across all lags for Model 2 with three fixed knots for the exposure variables. The grey area illustrates the 95%-CI (confidence interval) of the modelled relationship. The vertical lines in the figures present the 1, 2.5 and 97.5 and 99 percentiles of the daily mean distribution of the variable in question.

**Figure 6 ijerph-19-13398-f006:**
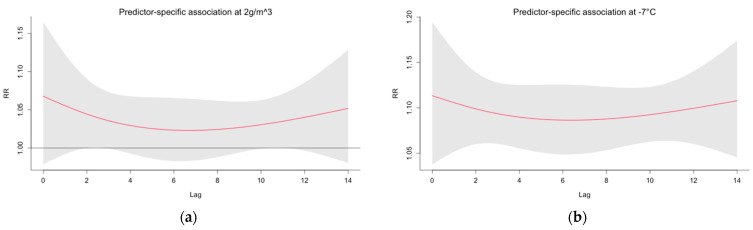
Relative risks (RR) of COVID-19 at (**a**) 2 g/m^3^ and (**b**) −7 °C over a 14-day lag, for the time period 1 August 2020 until 31 May 2021 at the Helsinki-Uusimaa Hospital district. The predictor-specific associations were calculated with Model 2 with three fixed knots for each predictor variable. The grey area illustrates the 95%-CI (confidence interval) of the modelled relationship.

**Table 1 ijerph-19-13398-t001:** An overview of COVID-19 cases and meteorological data during the two study periods. The total number of laboratory confirmed COVID-19 cases, range of the daily average absolute humidity, the range of the daily average temperature values and minimum and maximum temperatures for the two time periods are shown. Temperature and absolute humidity values were measured at the Kumpula weather station.

Time Period	Confirmed COVID-19 Cases	Average Daily Absolute Humidity (g/m^3^)	Average Daily Temperature, Max, Min Values (°C)
27 February 2020–31 May 2021	54,203	1.01–14.01	−18.8–+24.00 min = −21.8 max = +29.4
1 August 2020–31 May 2021	48,013	1.01–13.9	−18.8–+21.7 min = −21.8 max = +27.1

## Data Availability

The TROPOMI Surface UV product is available at https://nsdc.fmi.fi/data/data_s5puv.php (accessed on 3 May 2022). The OMI UVR data were downloaded from the NASA Aura validation center https://avdc.gsfc.nasa.gov/ (accessed on 3 May 2022).

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
