# Peer review of "Impact of Selected Meteorological Factors on COVID-19 Incidence in Southern Finland during 2020–2021"

_ijerph, 2022, doi:10.3390/ijerph192013398_

Round 1

Reviewer 1 Report

This study has explored the impact of meteorological factors mainly temperature and humidity on the incidence of COVID-19 in Southern Finland.  I have the following comments:

1. Because countermeasures to curb of transmission of COVID-19 are very important, any relationship without considering the broader background of countermeasures can be false. However, the author only used an index of government response as a controlling factor in model 2. We don't know what exactly this index measures, and we don't understand the change of this index along with countermeasures during the research period either.

2. In the "2.1 Data" section, the authors mentioned that "we analyzed the two time periods separately", then how come we can only find their results of only one period from August 2020 to May 2021? Why did they choose to analyze these two periods separately (respectively Feb 2020 to May 2021, and Aug 2020 to May 2021)?

3. Where is Fig.7? Did the authors mean Fig.6 by saying Fig.7?

4. The authors mentioned that a new delta (B.1.617) variant started to spread in Finland and led to increased incidence during 2021 summer. Since the main finding of this study is that a higher incidence was linked to low absolute humidity and low ambient temperature. What will happen if the authors include the 2021 summer data in the analysis? Will the main conclusions been changed?

5. You can use the 2021 next half year data in the Finland to verify your main findings.

6. The author may consider a through discussion of their main findings with numerous extant research working on the meteorological factors of the transmission of COVID-19.

Reviewer 2 Report

Haga et al. investigated the impact of few meteorological factors on COVID-19 incidence in some part of Southern Finland during 2020-2021. The findings seem very interesting and important. I would like ask about the following points to improve the manuscript.

1)    The title could be more specific such as, Impact of few meteorological factors on COVID-19 pandemic waves in Southern Finland during 2020-2021

2)    Explanation on COVID-19 incidence versus mereological factor is really complex for such a short-term study. The researchers may focus on effect of the selected meteorological factors on three specific pandemic waves. This can make the story easier to understand.

However, the manuscript has been written in an impressive manner and I appreciate authors for their work and recommend its Acceptance if addressed well with reviewers’ comments and/or suggestions.

Reviewer 3 Report

The manuscript is well-written and structured. However, the main question can be asked as regards the novelty of the current work and why so, environmental factors such as temperature and humidity were associated with COVID-19 incidence in prior published articles. Other minor comments have been marked on the attached file.    

Round 2

Reviewer 1 Report

The manuscript has been sufficiently improved.

Reviewer 2 Report

It can be accepted from my side. Congratulations to the authors.